Proceedings of the 6th Symposium on Advances in Approximate Bayesian Inference, 2024 1–15

# Bayesian Optimization for Crop Genetics with Scalable Probabilistic Models

**Ruhana Azam**[1,2]                                                        razam2@illinois.edu
**Sang T. Truong**[2]                                                      sttruong@cs.stanford.edu
**Samuel B. Fernandes**[1]                                                 samuelbf@uark.edu
**Andrew D.B. Leakey**[1]                                                  leakey@illinois.edu
**Alexander Lipka**[1]                                                     alipka@illinois.edu
**Mohammed El-Kebir**[1]                                                   melkebir@illinois.edu
**Sanmi Koyejo**[2]                                                        sanmi@stanford.edu

[1] *University of Illinois Urbana-Champaign,*
[2] *Stanford University*

## Abstract

An overarching goal of crop improvement is to select plants with desirable traits so that crops can provide sufficient food and nutrients for humanity in the face of climate change. To achieve such a goal, crop breeders utilize genomic prediction, in which that genome-wide DNA marker information is used to predict breeding values for desirable traits . Genomic prediction is complemented by advancements in high-throughput phenotyping, in robots and drones collect orders of magnitude higher amounts of trait information than in the past. Although such data are abundant and easy to collect, identifying the most biologically meaningful traits for use in genomic prediction is expensive. Bayesian optimization (BO) is a strong cost-effective solution to identify such meaningful traits. In this work, we quantified the performance of BO with a collection of acquisition function and surrogate models for identifying good proxies, in a set of +4 million proxies. We found that BO achieves comparable sample efficiency to random search while requiring significantly less computation. Despite traditional BO and random search techniques performing sufficiently well, both search techniques fail to leverage information from related tasks. To this end, we propose a pre-trained model as a transfer learning method. Using this benchmark, we conduct an extensive empirical study and demonstrate promising results on the transfer learning effect, highlighting a core design principle for developing more parsimonious optimization algorithms for crop improvement.

## 1. Introduction

Farmers are under immense pressure as the world's population, projected to hit 10 billion by 2050 (Searchinger et al., 2019), demands more food, while urbanization and industrialization decrease available farmland (Satterthwaite et al., 2010; Follmann et al., 2021). Crop improvement efforts aim to enhance crop yield, efficiency, and profitability while minimizing waste and environmental impact. This approach relies on data-driven decision-making systems, such as GPS and remote sensing (Delgado et al., 2019; Boursianis et al., 2022), to create a more sustainable, productive, and efficient agricultural system (Liaghat et al., 2010; Sishodia et al., 2020).

One method of harnessing technology, which can boost crop yield and strengthen resilience, is using genomic prediction models with traits obtained from high-throughput phenotyping for crop breeding. In such a model, the additive effects of hundreds of thousands of genome-wide markers are the explantory variables and one or more high-throughput phenotypes are the response variables. While high-throughput data can be collected cheaply, selecting which traits are best-suited for genomic prediction models is expensive, with naive approaches requiring a combinatorial search. To this end, Bayesian optimization (BO) is considered a strong cost-effective solution for faster search. BO (Žilinskas, 1975; Shahriari et al., 2016; Frazier, 2018; Garnett, 2022) has consistently been proven to work well across a range of applications: hyperparameter tuning of machine learning models (Snoek et al., 2012), material design (Kiyohara et al., 2016; Balachandran et al., 2016), and drug discovery (Gómez-Bombarelli et al., 2018; Griffiths and Hernández-Lobato, 2020). BO requires a surrogate model and an acquisition function. Here we consider Gaussian Processes (GPs).

This paper is focused on identifying wavelength ratios with the highest co-heritability with the desired trait, using BO as the algorithmic search procedure. By leveraging a sequential and adaptive search over wavelength ratios, the number of co-heritabilities calculated can be reduced. Further, we curate the first benchmark, which consists of more than four million proxies, to understand the performance of BO methods identifying proxies for genomic prediction. We observe that different traits can have similar co-heritability properties. To this end, we propose a pretrained model to transfer the knowledge learned from one set of traits and the corresponding wavelength spectrum to another set to reduce the search cost. We present encouraging results showing that transfer learning of good wavelength ratios is possible for different traits. The code for the experiments conducted in this paper is available on GitHub[1].

Our contributions are as follows:

- We curate a dataset of more than four million proxy measurements that can be used to understand the performance of BO algorithms in the context for crop genetics. This unprecedented dataset offers a comprehensive foundation for evaluating and enhancing the efficacy of BO techniques for use in genomic prediction of crops.

- We establish the first BO benchmark in the context of proxy search. This benchmark, derived from our comprehensive dataset, provides a standard metric for assessing the efficacy of both traditional and novel BO approaches. It facilitates the comparative analysis of methods and drives the development of more efficient and effective optimization techniques tailored utilizing high-throughput data to make the best possible breeding decisions.

- We conduct an extensive empirical study demonstrating the significant impact of transfer learning on BO's performance in find proxy phenotypes for genomic prediction. Our findings demonstrate a promising transfer effect that paves the way for efficient optimization with minimal sample efficiency.

- We propose a novel application of transfer learning to BO by introducing a pretrained model approach. This innovation addresses the scalability issues inherent in traditional Gaussian Process (GP) surrogates by effectively leveraging information from

---

1. https://github.com/stair-lab/bo4ag

related tasks. Our method enhances BO's capability to identify valuable proxies with reduced computational expense and improved optimization efficiency.

## 1.1. Background on the scientific motivation

One method of harnessing technology, which can boost crop yield and strengthen resilience, is incorporating genomic prediction models for crop breeding. These models enable biologists to predict the genomic estimated breeding values (GEBVs) of targeted traits using genome-wide markers of plants. Genomic prediction enables biologists to identify plants with desirable traits early in the growing season, eliminating the need to wait for trait observation. Collecting data for these predictive models is costly, either requiring teams of scientists to go out into fields to collect data or to bring individual plants into labs (Gastal et al., 2015; Berger et al., 2020; Jurado et al., 2022). To reduce costs, high-throughput phenotyping data are used in place of directly measuring the trait of interest. For example, scientists may favor plants with better water absorption. It is known that the degree of water absorption is correlated with the wavelength spectrum of crops (Roberts et al., 2018). Hence, an efficient way to collect data is to fly a drone mounted with a hyperspectral imaging sensor to measure the wavelengths emitted from each crop (Maimaitijiang et al., 2020; Jin et al., 2020).

Measuring the usefulness of wavelength spectra for use in genomic prediction models hinges on two factors (Janssens, 1979; Fernandes et al., 2023): (i) the correlation between the wavelengths and the desired trait, $r(w, t)$, and (ii) the amount of variance in the wavelength(s) $w$ and the trait $t$ of the crop cohorts attributed to genetics, $h_w^2$ and $h_t^2$, respectively (Falconer and Mackay, 1983). These factors can be amalgamated into a metric called "co-heritability." Discovering the relationship between wavelength spectra to co-heritability is non-trivial because it is unique to the cohort of crops (e.g. genetics and species dependent). Moreover, a brute force search becomes increasingly impractical as the number of proxy traits increase (Fernandes et al., 2023; Azam et al., 2023). To address the expense associated with sorting through various wavelengths, we take the Bayesian optimization (BO) approach, which can effectively reduce the number of wavelengths (s) considered.

## 2. Preliminaries

For a more comprehensive understanding, we'll begin by introducing several essential concepts integral our problem:

**Co-heritability** For a given trait $t \in \mathcal{T}$, heritability ($\mathbf{h}_t^2 : \mathcal{T} \to [0, 1]$) is a crucial metric that delineates the extent to which genetic factors account for the variability observed, as opposed to environmental influences (Falconer and Mackay, 1983). Mathematically, $\mathbf{h}_t^2 = \frac{\sigma_{t,g}}{\sigma_{t,g} + \sigma_{t,e} + \sigma_{t,r}}$, where $\sigma_{t,g}^2, \sigma_{t,e}^2, \sigma_{t,r}^2$ represents the genetic variance, environmental variance, and residual variance inherent in a trait $t$, respectively. For plant breeders, traits with high heritability are ideal because the offspring is likely to inherit its parents' traits. Co-heritability ($\mathbf{coh} : \mathcal{T} \times \mathcal{T} \to \mathbb{R}$) is a measure that combines the heritability of two traits and their Pearson correlation $r$ between the two traits (Janssens, 1979; Fernandes et al., 2023): $\mathbf{coh}^2(t_1, t_2) = \sqrt{\mathbf{h}_{t_1}^2} \times \sqrt{\mathbf{h}_{t_2}^2} \times r(t_1, t_2)$. To estimate the Pearson correlation, we build a new trait, which is the sum of $t_1$ and $t_2$. Using this new trait, we estimate the

Pearson correlation: $r(t_1, t_2) \approx \frac{\sigma^2_{(t_1+t_2)}}{\sigma^2_{t_1}\sigma^2_{t_2}}$. By calculating the co-heritability of a desired trait with a proxy trait, we can quantify the goodness of utilizing the proxy for a task of genomic prediction. One major detail has been left out of these definitions so far – how the variance terms (e.g. $\sigma^2_{t,g}, \sigma^2_{t,e}, \sigma^2_{t,r}$) are calculated? The relationship between genetic and environmental effects of a trait is typically modeled using a linear mixed-effect model (LMM) (Speed et al., 2012; Mrode, 2014).

**Linear Mixed-effect Models** Linear mixed models (LMMs) are among the most effective statistical tools for modeling the influence of environmental and genetic factors on an observed trait (Piepho and Mohring, 2007; Holland et al., 2003; Speed et al., 2012). LMMs use a hierarchical structure to differentiate between fixed effects, which represent population-level influences, and random effects, which capture group-specific deviations (Bates et al., 2014; Butler et al., 2017). The observed variable for $N$ individuals, $\mathbf{Y} \in \mathbb{R}^N$, has the following data generating process: $\mathbf{Y}|u \sim \mathcal{N}(\mathbf{Z}\Lambda u + \mathbf{X}\beta, \sigma^2\mathbf{I}_N)$ where $u \sim \mathcal{N}(0, \sigma^2\mathbf{I}_Q)$. The fixed effect is $\beta \in \mathbb{R}^P$, where $P$ is the number of fixed effects and $\mathbf{X} \in \mathbb{R}^{N \times P}$ is the design matrix corresponding to the fix-effect. $\beta$ is often used to capture population structure (i.e. influences shared among a large portion of the population) (Bates, 2011). The random effect is $\Lambda u$, where $\Lambda$ is a block diagonal matrix composed of covariance matrices $\Lambda = \mathrm{diag}(\Lambda_1, ..., \Lambda_k)$; one covariance matrix per grouping factor. Random effects can represent smaller individual or group-level variations within a population (i.e., proportional to an additive genetic relatedness matrix, calculated from genome-wide markers (VanRaden, 2008)). The corresponding design matrix, $\mathbf{Z} \in \mathbb{R}^{N \times Q}$, is composed of $k$ vectors for each grouping factor: $\mathbf{Z} = [z_1, ..., z_k]$.

LMMs offer a robust approach to quantifying genetic and environmental effects in plant populations while accounting for potential sources of variation and relatedness among individuals. So, how do we model traits? With $Y$ corresponding to the observed trait, $X$ captures the crop's overall genetic effect. Finally, the matrix $Z$ is built using grouping factors, based on the crop's genetics or environments (e.g. location). This allows us to model the genetics and environmental variation (i.e. the diagonal of $\Lambda$). In practice, LMMs are fitted using the restricted maximum likelihood objective. We refer the reader to a tutorial of the `lmer4` R package (Bates et al., 2014), which gives an in-depth understanding of the objectives and the numerical methods used to fit such models. With a basic understanding of how LLMs can be used to model the genetic and environmental variances effect on a trait, we have a complete understanding of how the (co)-heritability of traits can be calculated.

**Bayesian Optimization** Domain experts have identified co-heritability between a proxy trait and a desired trait as a good way to quantify how well a proxy may work in genomic prediction models. This paper aims to detect which wavelength ratio, $\frac{w_1}{w_2}$, can act as the best proxy for a desired trait $t$: $w_1^*, w_2^* = \arg\max_{w_1, w_2 \in W} \mathbf{coh}^2(\frac{w_1}{w_2}, t)$, unfortunately derivative over the co-heritability function is non-trivial. Furthermore, the computation of each co-heritability value demands $\sim 0.2$ seconds, a potentially sluggish process contingent upon the granularity and scale of the hyperspectral data. Given these challenges, treating co-heritability as a black box within a Bayesian optimization (BO) framework becomes a pragmatic approach.

BO (Frazier, 2018; Garnett, 2022; Shahriari et al., 2016) is a technique for the global optimization of expensive, black-box functions. Such methods aim to solve the following optimization problem: $x^* = \arg\max_{x \in X} f(x)$. The black-box function, $f$, is one where you do not have access to its derivative. BO utilizes a probabilistic surrogate model to estimate $f$, then adaptively selects the next data point, ensuring that we extract the maximum information from each experiment. BO has consistently been proven to work well in a variety of benchmarks: hyperparameter tuning of machine learning models (Snoek et al., 2012), material design (Kiyohara et al., 2016; Balachandran et al., 2016), drug discovery (Negoescu et al., 2011; Shields et al., 2021), and genomic prediction (Tanaka and Iwata, 2018). BO requires a surrogate model and an acquisition function GP Beyond being proven to be a reliable method, GPs support exact inference, are interpretable and have straightforward uncertainty quantification, often making GPs with Matérn kernels the default choice of surrogate function in BO (Rasmussen and Williams, 2006). In this work, we wish to find the spectra with the optimal co-heritability.

**Probabilistic Surrogate Models** The Gaussian process (GP) is a powerful tool for probabilistic modeling commonly used in BO, offering the benefits of flexibility and a principled approach to uncertainty quantification. However, its scalability to large datasets has been a significant limitation, primarily due to the $\mathcal{O}(n^3)$ computational complexity associated with inverting the covariance matrix, where $n$ is the number of data points. This scalability issue has been a major barrier as the computational resources required to handle large-scale data are prohibitive Rasmussen and Williams (2006).

Recent advances in machine learning technology have begun to address these challenges. Stochastic Dual Descent (SDD) and Probabilistic Transformers present promising avenues for improving the scalability of GP regression (Lin et al., 2023). SDD offers a way to approximate the inverse of large matrices more efficiently, reducing the computational burden. On the other hand, Probabilistic Transformers leverage deep learning architectures to model distributions over functions, potentially bypassing the need for explicit covariance matrix inversions and enabling GP-like inference on much larger datasets (Wang and Ribeiro, 2020). These technologies herald a new era for GP regression, where its application to large-scale data and transfer learning becomes increasingly feasible. Transfer learning is a technique where a model trained on one task is repurposed or fine-tuned for another related task. It is beneficial if data is expensive or scarce, as it leverages knowledge gained from previous task(s) to improve performance on the target task. In our application of co-heritability search, as seen in Figure 6, we observe the similarities between the co-heritabilities spaces for four target traits. We wish to leverage such similarities.

## 3. Experimental Setup

Our dataset comprises 869 Sorghum Lines from two growouts near the University of Illinois Urbana-Champaign (Dos Santos et al., 2020; Ferguson et al., 2021). In our experiment, we consider four target traits 1) Nitrogren Area (narea) 2) Specific Leaf area (sla) 3) PLSR Nitrogren Area (pn) 4) PLSR Specific Lead area (ps). For each crop, the data consists of spectrography ranging from 350nm–2500nm. In this work, the proxy trait consists of wavelength ratios $(w_1/w_2)$ spectrography ranging from 350nm–2500nm. This is because wavelength ratios are commonly known to be a useful proxy (Carter, 1994; Read et al.,

2002; Roberts et al., 2018; Lu et al., 2018), and the space is small enough to illustrate. Note that proxies can be constructed with more complex linear combinations of functions, e.g., $\frac{w_1+w_2}{w_3+w_4}$).

In the experiments presented in Section 4.1 benchmark BO with GPs. We test Matérn kernels with $v = 1/2, 3/2, 5/2$ and radial basis kernels with adaptive length scales. The length scale is assigned a Gamma(3.0, 6.0) prior to the input and a Gamma(2.0, 0.15) prior to the output scale. Three acquisition functions were tested: 1) Expected Improvement (BO-EI) 2) Upper Confidence Bound (BO-UCB) and 3) Probability Improvement (PI). BO-UCB was executed with varying exploration levels $\beta = 0.1, 0.2, 0.5$. A uniform random search served as a baseline. Each experiment comprised five trials, beginning with 10 initial points. Queries were made for 300 iterations.

In Section 4.2, we present evidence that pretraining a GP model for multiple tasks improves the performance of co-heritability search on similar tasks. To illustrate this point, we perform (vanilla) GP regression on 500–3000 data points. At every step, points are uniformly sampled from each task. The validation set comprises 1000 uniformly sampled points, distinct from the training set. The GP employs a mixed kernel. This kernel combines Matèrn-kernel(s) for the continuous variable $(w_i)$, i.e., wavelength pairs and a categorical kernel(s) for discrete variables $(t_i)$, i.e., the trait:

$$K((\mathbf{w}_1,\mathbf{t}_1),(\mathbf{w}_2,\mathbf{t}_2)) = K_{\mathrm{mat}_1}(\mathbf{w}_1,\mathbf{w}_2) + K_{\mathrm{cat}_1}(\mathbf{t}_1,\mathbf{t}_2) + K_{\mathrm{mat}_2}(\mathbf{w}_1,\mathbf{w}_2) \cdot K_{\mathrm{cat}_2}(\mathbf{t}_1,\mathbf{t}_2). \quad (1)$$

Experiments conducted in Sections 4.1 and 4.2, both consist of GPs' with multiple hyperparameters (i.e. noise level, kernel's length scale). To select a good set of hyperparameters, we train a Vanilla GP on a smaller training set (i.e., 3000 points randomly sampled from all tasks) where the hyperparameters are tuned by minimizing the marginal log-likelihood using the default L-BFGS-B minimizer in sciPy. We observe that, regardless of the optimizer's initialization, the hyperparameters are shown to consistently converge to the same values.

## 4. Results and Discussion

### 4.1. Bayesian Optimization without Transfered Learning

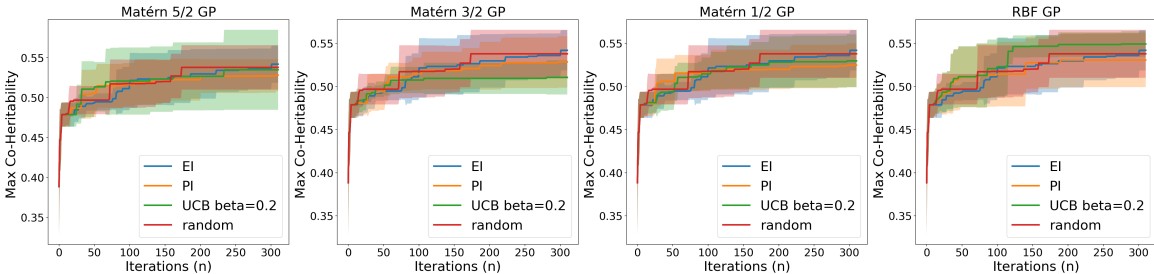

Figure 1: Performance (max co-heritability for narea) vs. number of iterations for BO for each co-heritability search. Each experiment was run for 300 iterations.

In this section, we present benchmarks for utilizing Bayesian Optimization with Gaussian Processes to identify high co-heritability wavelength ratios. As depicted in Figures

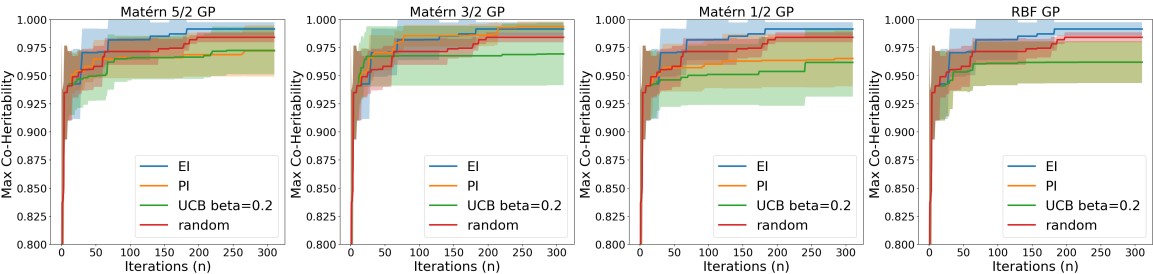

Figure 2: Performance (max co-heritability for pn) vs. number of iterations for BO for each co-heritability search. Each experiment was run for 300 iterations.

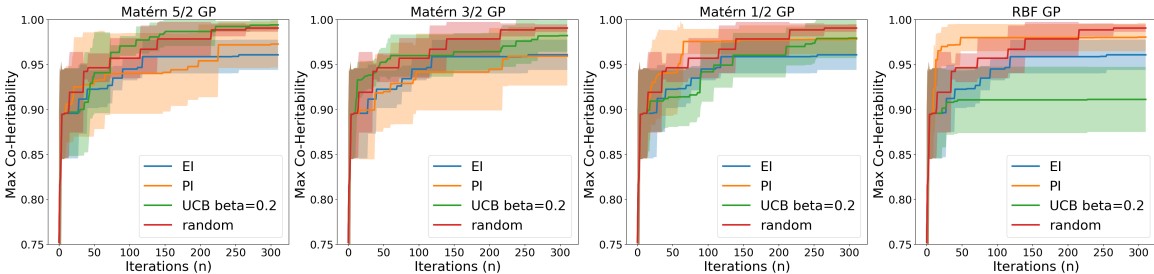

Figure 3: Performance (max co-heritability for sla) vs. number of iterations for BO for each co-heritability search. Each experiment was run for 300 iterations.

1, 3, 2 4, we benchmark three acquisition functions and 5 kernels over four different trait spaces. Our results indicate similar behavior over all four traits. With some exceptions, random search always performs within the standard deviation bound of BO after 300 iterations. In the other cases, like BO-UCB spectral-10 for ps, random search outperforms BO. This indicates that GP-based BO does not meaningfully outperform random search for co-heritability search over wavelength ratio. Considering that the runtime of random search is $O(1)$ per iteration while fitting a Gaussian process takes $O(n^2)$, it becomes evident that the cost-performance trade-off does not warrant the use of GP-based Bayesian optimization despite its marginal performance advantage.

Our result is robust to the choice of hyperparameters of the acquisition function. As a demonstration, we look at the effects of changing the exploration-exploitation levels in the

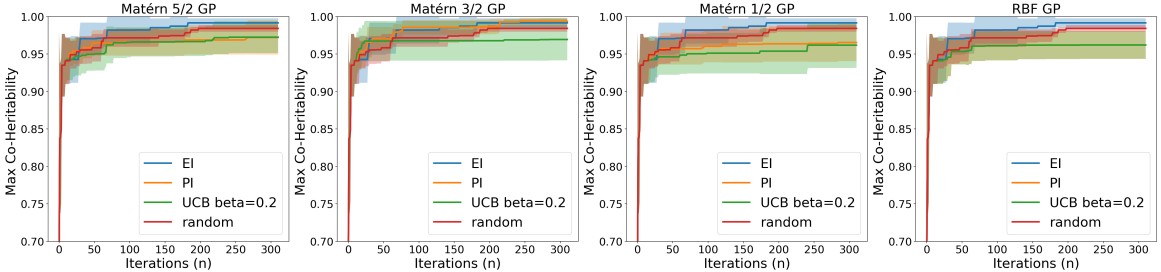

Figure 4: Performance (max co-heritability for ps) vs. number of iterations for BO for each co-heritability search. Each experiment was run for 300 iterations.

UCB objective on the performance of BO-GPs for search over co-heritabilities of wavelength ratios. To test the hyperparameter sensitivity, we execute the methods with different $\beta$ values, $0.1, 0.2, 0.5$. The results, presented in Figure 5, indicate that random search performs better or only marginally worse than any BO-UCB, regardless of the hyperparameter. We conclude that BO-GP with UCB is stable to the tuning $\beta$ for searching over hyperspectral co-heritability spaces. To the best of our knowledge, regardless of the acquisition function, BO with GPs for co-heritability search over wavelength ratios, surprisingly, achieves comparable performance to a simple random strategy while requiring significantly less computing cost.

Our experiments show that using Gaussian processes-based Bayesian optimization for finding high co-heritabilities may not perform much better than random search. We hypothesize several characteristics leading to the small gap in performance between GP-based Bayesian optimization and random search. First is sharpness – there are large smooth regions with little to no signal towards the optimal co-heritability. Regarding Euclidean distance, as seen in Figure 6, the top 1% of points are extremely close to points with the lowest heritability values, making these regions extremely sharp. Since Gaussian processes with Matérn kernels are most effective for representing smooth functions, they are unlikely to represent extremely sharp regions accurately. Second is aperiodicity, the aforementioned patterns are repeated throughout the search space, yet in a non-periodic manner. Despite finding that GP-based BO performs similarly to random search, in the next section, we propose one potential direction for boosting the performance BO for co-heritability search.

## 4.2. Transfer Learning Improves Heritability Prediction

Transfer learning proves especially advantageous for tasks with similar characteristics, allowing the utilization of knowledge acquired from one task to enhance performance on related tasks. This approach has the potential to enhance predictive accuracy and efficiency in our context (Kendall et al., 2018; Standley et al., 2020; Bai et al., 2023). As illustrated in Figure 6 the search space of co-heritabilities for various phenotypes, reveals multiple overlapping peaks and plateaus. In this section, we leverage the similarities between different co-heritability search spaces. Our objective is to develop a multi-task model capable of predicting the co-heritability of wavelength ratios across four distinct tasks simultaneously. Through this experiment, we aim to showcase the effectiveness of positive transfer in forecasting the co-heritability of wavelength ratios.

As illustrated in Figure 7, with 3000 data points, a consistent trend emerges: the validation loss steadily decreases across all tasks as the dataset size grows for each task. This reduction in mean squared error indicates that, on average, the training data from various tasks jointly improve the performance of all tasks. Furthermore, as depicted in Figure 6 (bottom), we present the GP model acquired after training on $N = 3000$ points. The resemblance of these plots to those in Figure 6 (top) assures that a robust GP model is being estimated. These findings affirm the potential of leveraging larger, diversified datasets to refine and strengthen GP models, thereby enhancing their applicability and effectiveness in complex multi-task scenarios.

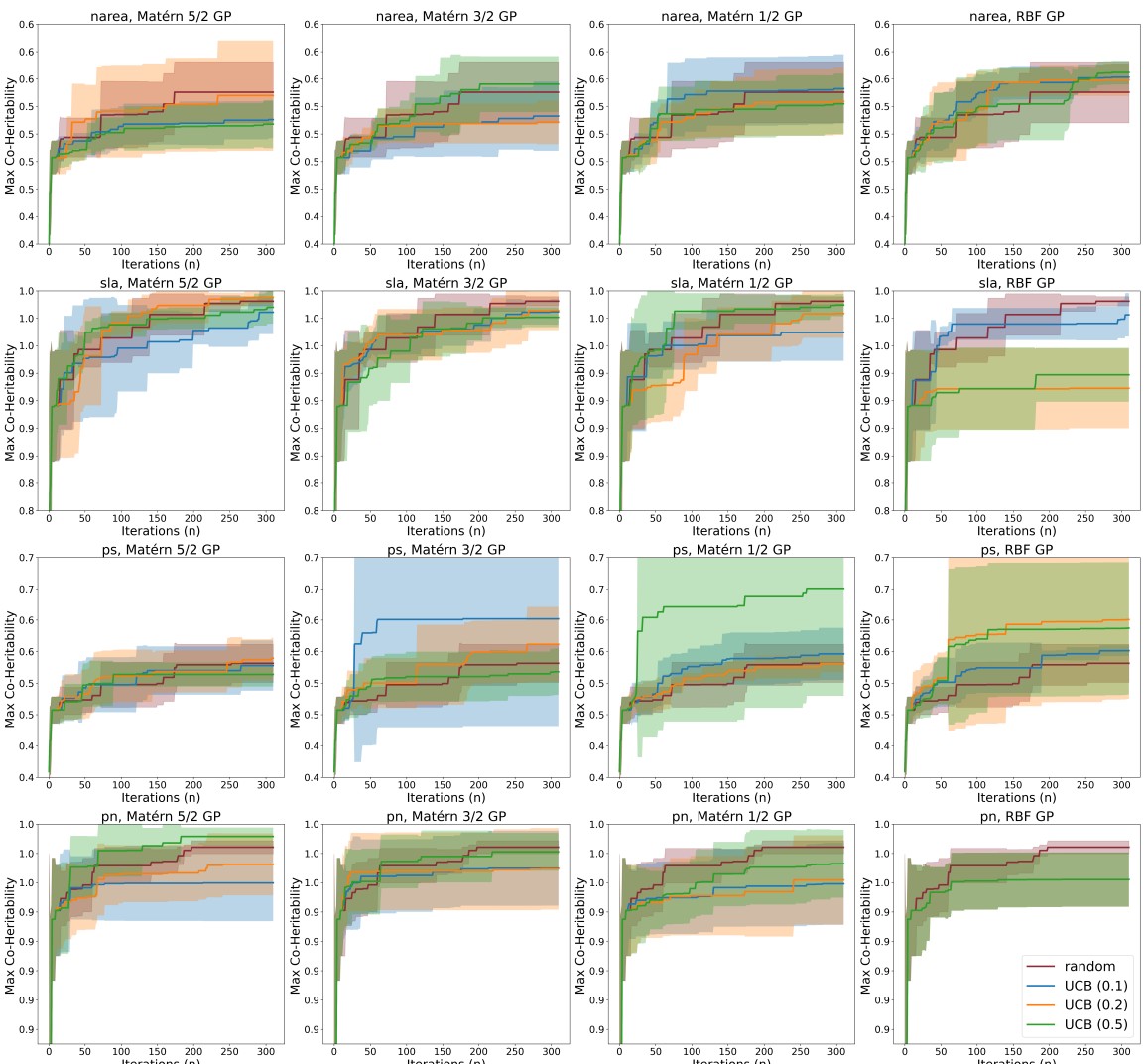

Figure 5: Performance (mean and standard deviation bands) vs. time to search over different co-heritability settings for BO-UCB. The method was run over different levels of exploration-exploitation, $\beta = 0.1, 0.2, 0.5$.

## 5. Conclusion & Future Work

We conduct an extensive empirical study in the effectiveness of using BO methods in the search of good proxies for genomic selection. Our results indicate that Bayesian optimization (BO) attains performance on par with random search, yet demands substantially fewer computational resources. In the second part of this study, we assess the posterior predictive distribution based on data from four distinct phenotypes. Our initial findings indicate a promising transfer effect in learning the co-heritabilities of various target traits for Sorghum, using wavelength ratios as indicators for genomic prediction models. This indicates that pretrained models are a promising direction for efficiently doing proxy search for a cohort

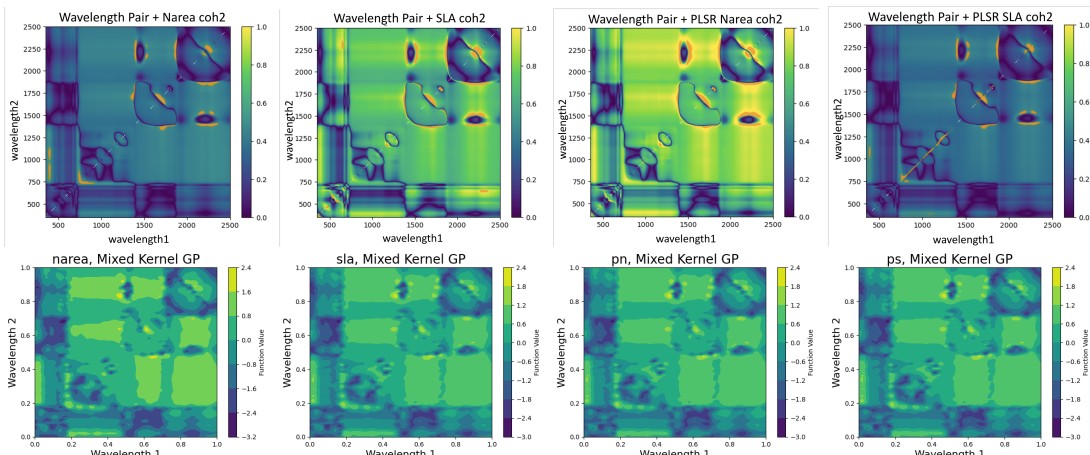

Figure 6: First row: An illustration of the search space of co-heritabilities of nitrogen area, specific leaf area, plsr nitrogren area, and plsr specific leaf area (left to right) for each wavelength ratio $\frac{w_1}{w_2}$. The orange highlights the top-1% of the search space in the crop Sorghum. Second row: Multitask GP posteriors after training on $N = 3000$ co-heritabilities points nitrogen area, specific leaf area, plsr nitrogren area, and plsr specific leaf area (left to right).

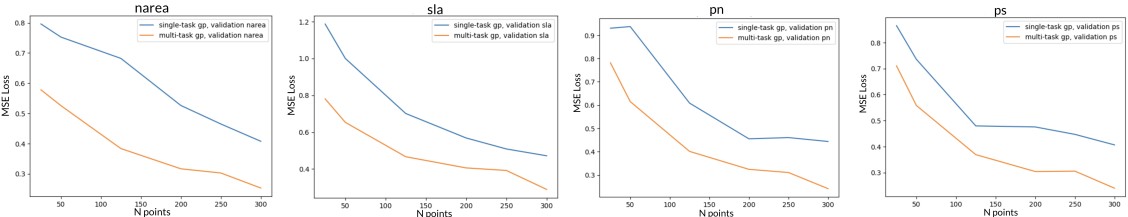

Figure 7: Vanilla GP trained on four target traits with increasing training points.

of plants. However, a more thorough examination is required to fully confirm this effect and its transferability between specific phenotypes.

Our experiments involved $3,000$ data points. However, GPs suffer from high computational costs, with time complexity of $(O(n^3))$ and space complexity of $(O(n^2))$. The high computational costs of GPs can limit the number of points the pretrained model is able to capture. This becomes an issue as pretraining on 4 phenotypes can potentially be composed of up to 18.5 million wavelength ratios. As future work, to address the scalability issue, we intend to utilize a stochastic gradient method known as Stochastic Dual Descent (SDD) to streamline the fitting process. SDD has been proven to efficiently estimate kernel regression parameters through stochastic gradient descent by optimizing the dual objective. This approach necessitates domain expertise to select appropriate kernel hyperparameters. We hypothesize that an optimal hyperparameter, determined through Maximum Likelihood Estimation, will remain effective as the volume of training data increases. If hyperparameters show to be stable, we will utilize these hyperparameters to decide on a kernel for the SDD method.

To the best of our knowledge, we are the first to study the effects of using Bayesian optimization as a method finding the most biologically meaningful subset of traits from high-throughput phentoyping data. This paper proposes using pretrained models as a transfer learning approach for Bayesian optimization to identify valuable low-cost proxies for predicting desirable plant traits in precision agriculture, demonstrating promising results in leveraging information across related tasks. Our study addresses a way to reduce the costly data collection process for genomic prediction, enabling more efficient and effective manners to breed crops for the future.

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

## Appendix A. Fitting the Surrogate Model

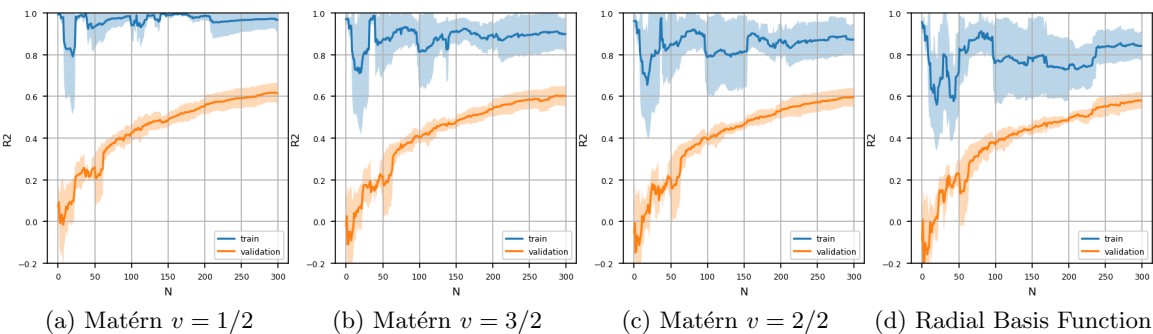

(a) Matérn $v = 1/2$     (b) Matérn $v = 3/2$     (c) Matérn $v = 2/2$     (d) Radial Basis Function

Figure 8: The $R^2$ of various Matérn kerels which have Gamma(3,6) for the lengthscale prior and Gamma(2, 0.15) output scale prior.

In an attempt to better capture the global trends and the small local trends of this search space, we use an additive kernel. The additive kernel is composted of a radial basis function and Matérn $v = 1/2$ kernel. All kernels use a Gamma(3,6) lengthscale prior and Gamma(2, 0.15) output scale prior.

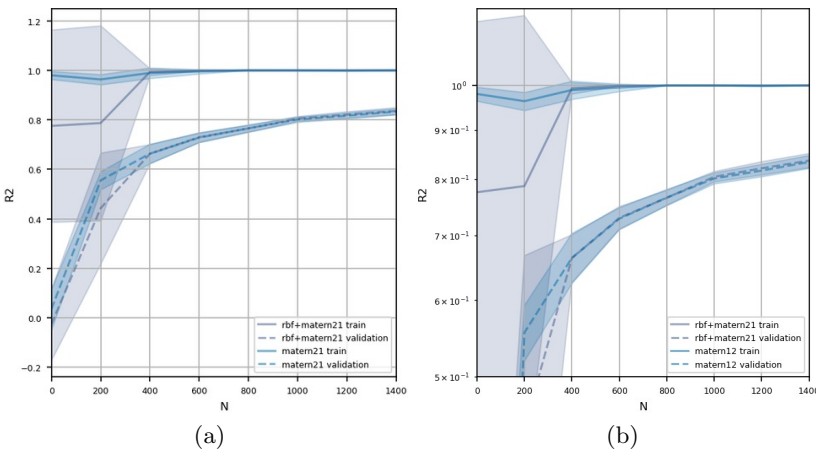

(a)            (b)

Figure 9: The $R^2$ comparison between an additive kernel, Radial Basis Function + Matérn $v = 1/2$ and a Matérn $v = 1/2$ kernel. The right figure us shown in log scale.

Fyi

As seen in Figure 9, the additive kernel marginally outperforms the Matérn $v = 1/2$ kernel in the last 600 iterations. We suspect that the additive kernel performs better only when a sufficient number of samples are provided to gain a signal of the sharp regions of the search space. Due to this reason, the additive kernel performs better when more samples are provided.

