# OpenReview forum: "Bayesian Optimization for Crop Genetics with Scalable Probabilistic Models"
_approximateinference.org/AABI/2024/Symposium_Archival_Track — AABI 2024 - Archival Track_

### Official Review · Reviewer_1SNE · 2024-04-20

**Rating:** 6
**Confidence:** 3

**Review:**

This paper studies the usage of Bayesian optimization (BO) in an agricultural setting. Specifically, it is used to find new novel variants of plants that have desirable traits such as climate-change resistance. Experiments show that BO (with GPs) often does not provide significant gains over very simple random search.

**Significance**

The paper provides a critical take on the usage of BO-GP in a very much real-world setting. This kind of discourse is very important to drive the search for better algorithms that can solve real-world problems well. Unfortunately, the authors only discuss the potential "fix" for BO-GP in a vague way.  Maybe the authors could take inspiration from different fields where BO on top of some pretrained embeddings is the norm---see [1], [2], [3] for examples. I would guess that this can help mitigates the issues mentioned in Fig. 6 where the search space is very difficult to navigate (by contrast, the embedding space should be nicer).

**Originality**

I am not familiar with the agricultural setting that this paper focuses on. However, it does seem that this critical take on BO-GP is novel. Indeed, this is a breath of fresh air since in many fields, BO-GP is the _de facto_ method that is assumed to always work out-of-the-box.

**Quality and clarity**

This is the place where the paper could be improved further.

1. The authors only study a single dataset with 869 candidates. It would make the paper's results stronger if the authors considered more datasets with varying amounts of candidates to optimize. After all, this is a benchmark paper.
2. While the writing is good, the exposition of the results in the figures can be improved significantly. For example, Figs. 1-4 can each be made into a single-row figure (the random search baseline is always the same, so duplications into 3 rows in the figure are wasteful). Moreover, the fonts are so tiny and impact readability. I suggest the authors use this package to (automatically!) get the correct font size for their figures: https://github.com/pnkraemer/tueplots.

**Conclusion**

I think the message of this paper---a critical look at GP-BO in a real-world setting; it's not better than random search!---is interesting and important. However, there are some missing discussions (e.g. what would be the concrete fix?)


**References**

[1] https://chemrxiv.org/engage/chemrxiv/article-details/6509b003ed7d0eccc3d2f70f

[2] https://arxiv.org/abs/2402.05015

[3] https://arxiv.org/abs/2203.12742

---

### Official Review · Reviewer_8iRq · 2024-04-24
**Clever idea and good contribution**

**Rating:** 7
**Confidence:** 4

**Review:**

Quality and Clarity
---
The paper focuses on the observation that different wavelength ratios in genetic data have similar co-heritability properties which enables transfer learning to be applied, thereby reducing computation overhead. In addition, a benchmark for comparative analysis of different BO approaches is provided.

Description of the related preliminaries, such as co-heritability, linear mixed-effect models (LMM), and stochastic dual descent (SDD) is clearly given. In addition, the strengths and short-comings of each works is provided which leads to the proposed idea of employing transfer learning to mitigate the computation complexity while maintaining the quality of the prediction.

The environment for experiments, along with the hyper parameters are also clearly defined to support reproducibility. The proposed benchmark does exhibit and expose interesting characteristics of different surrogate models.


Originality and Significance
---
Finding a way to manageably apply BO to genetics without the need for excessive amount of computation by reducing the search space will benefit precision agriculture. The paper's significance is shown through identifying correlation between co-heritability of different traits or proxies and utilising this observation to facilitate transfer learning. The combination of proposed benchmark and dataset seems to be extensible which could be a foundation for future research.


Pros
---
- Narrative includes abundant description and information about related knowledge.
- The authors stress on expandability of the proposed idea on to larger dataset and points out important features of performance results.

Cons
---
- The paper seems to have minor grammar errors and typos.
- The letters in the figures are too small to be legible.

---

### Official Review · Reviewer_rYLV · 2024-04-24
**Review of "Bayesian Optimization for Precision Agriculture with Scalable Probabilistic Models"**

**Rating:** 5
**Confidence:** 3

**Review:**

The paper studies the application of Bayesian optimization in the selection of proxies in precision agriculture. The goal is to select a good set of proxies from more than four million proxies to help genomic selection of plants with desirable traits. The paper proposes this new dataset with extensive numerical experiments, compares different BO methods with random search, and extends further to transfer learning using pretrained models.

Strength:

The application of BO in precision agriculture is new and interesting. The numerical experiments are extensive.

Weakness:

The exposition is not clear for readers to understand how BO is applied to the concrete problem. There is a severe lack of math/technical details and necessary descriptions of the dataset. The description of linear mixed-effect models seems out of place. As a result, the entire application is quite confusing and makes it hard to understand how BO is applied in the context.

1. The linear mixed-effect model is introduced on page 3-4. However, it is never mentioned anywhere again in the rest of the paper. Given the technical details for LMM on page 3-4, it seems LMM is important for the data application. Therefore, it is confusing what is the purpose of introducing LMM.

2. If the main purpose of LMM is to introduce the variance components in the definition of $h_t^2$, then the other two important quantities, $h_w^2$ and $r(w,t)$, have no mathematical definition in the paper. These two quantities are crucial for properly defining the final response variable, the co-heritability $f(w, t)$.

3. The paper does not give much description of the data structure. From the beginning of Section 3, we only know there are 869 Sorghum Lines from two growouts. However, at the beginning of the paper, it is mentioned that the BO is applied to select from a set of more than 4 million proxies. So where do the 4 million proxies come from? Are they included in the dataset?

4. It is not clear how BO is applied to the dataset:

4(i) What is the data model? What are the variables to optimize over in BO?

4(ii) At the beginning of Section 3, it says that "we consider four target traits". But I think the target should be the co-heritability? Note that in all the figures, the vertical axes are "Max Co-Heritability".

4(iii) In the 3rd and 4th paragraphs of Section 3, it is mentioned that GP regression was fitted on 500–3000 data points. I do not know which variables are associated with these data points. Are they the data points from the original dataset, or some points in the space of variables to optimize over in BO?

---

### Official Review · Reviewer_iacF · 2024-04-25
**BO on precision agriculture**

**Rating:** 4
**Confidence:** 4

**Review:**

This paper quantifies the performance of BO on the application of precision agriculture, which identifies good proxies among 4M+ proxies. In order to leverage information from related tasks, the authors propose a pre-trained model as a transfer learning method. Experiments in co-heritability search on four target traits showed promising results of BO when utilizing transfer learning compared to random baseline.



Pros:

* The paper is well written and easy to follow
* The paper tackles an interesting problem and can potentially be impactful in practice



Cons:

* Section 4.1 on the experimental results of BO showed negative results compared to random search; section 4.2 claims the BO with transfer learning improves the performance, however, it is comparing the single-task GP with multi-task GP and lacks other baseline methods for comparison. Overall, it is not clear how the paper shows benefits of approximate Bayesian methods on the application of precision agriculture.

* The novelty of the paper is limited. There is no new method proposed and is mainly an application of existing BO techniques in the specific application of precision agriculture.

* Figure 1-4 in the experimental section seems a bit repetitive. Presumably the baseline random search is the same across the 3 acquisition functions and kernels; better presentation of the results can make the paper more compact and convey only the most important information.

---

### Meta-Review · Area_Chair_chEL · 2024-05-20

**Recommendation:** Accept (Poster)
**Confidence:** 3

**Metareview:**

The paper investigates the application of Bayesian Optimization (BO) in precision agriculture, focusing on identifying effective proxies for genomic selection among over four million potential proxies. The study introduces a pre-trained model leveraging transfer learning to enhance BO performance. The reviewers provide a range of feedback highlighting the paper’s strengths and areas for improvement.

Overall, the paper is well written and easy to follow and tackles a problem of significant practical and societal relevance. However, apart from the application, there is limited methodological novelty as well as unfortunately weak experimental results. The authors argue that pretraining on multiple tasks (multi-task learning in Fig 7) aids the performance but the numerical results indicate no clear benefit and also the proposed GP approach is not competitive with a random acquisition. While the results are not positive, the authors curate a new data set for this interesting real-world task. This could be valuable to the BO community, who can take on the challenge of improving over the random acquisition baseline.

I urge the authors to improve the experimental description and text according to the reviewers' feedback. Also, it would be beneficial to provide code for using the curated data set so it can actually be used for further BO developments. Trusting the authors that they will make the necessary changes, I recommend acceptance.

---

### Decision · Program_Chairs · 2024-05-27

Accept